# Odorant Binding Proteins (OBPs) and Odorant Receptors (ORs) of *Anopheles stephensi*: Identification and comparative insights

**Zeeshan Zafar**[1], **Sidra Fatima**[1], **Muhammad Faraz Bhatti**[1]*, **Farooq A. Shah**[2], **Zack Saud**[3], **Tariq M. Butt**[3]*

1 Atta-ur-Rahman School of Applied Biosciences (ASAB), National University of Sciences and Technology (NUST), Sector H-12, Islamabad, Pakistan, 2 Razbio Limited, Bridgend, United Kingdom, 3 Department of Biological Sciences, Swansea University, Swansea, United Kingdom

* mfbhatti@asab.nust.edu.pk (MFB); t.butt@swansea.ac.uk (TMB)

**Data Availability Statement:** All relevant data are within the manuscript and its Supporting Information files.

## Abstract

*Anopheles stephensi* is an important vector of malaria in the South Asia, the Middle East, and Eastern Africa. The olfactory system of *An. stephensi* plays an important role in host-seeking, oviposition, and feeding. Odorant binding proteins (OBPs) are globular proteins that play a pivotal role in insect olfaction by transporting semiochemicals through the sensillum lymph to odorant receptors (ORs). Custom motifs designed from annotated OBPs of *Aedes aegypti*, *Drosophila melanogaster*, and *Anopheles gambiae* were used for the identification of putative OBPs from protein sequences of the *An. stephensi* Indian strain. Further, BLASTp was also performed to identify missing OBPs and ORs. Subsequently, the presence of domains common to OBPs was confirmed. Identified OBPs were further classified into three sub-classes. Phylogenetic and syntenic analyses were carried out to find homology, and thus the evolutionary relationship between *An. stephensi* OBPs and ORs with those of *An. gambiae*, *Ae. aegypti* and *D. melanogaster*. Gene structure and physicochemical properties of the OBPs and ORs were also predicted. A total of 44 OBPs and 45 ORs were predicted from the protein sequences of *An. stephensi*. OBPs were further classified into the classic (27), atypical (10) and plus-C (7) OBP subclasses. The phylogeny revealed close relationship of *An. stephensi* OBPs and ORs with *An. gambiae* homologs whereas only five OBPs and two ORs of *An. stephensi* were related to *Ae. aegypti* OBPs and ORs, respectively. However, *D. melanogaster* OBPs and ORs were distantly rooted. Synteny analyses showed the presence of collinear block between the OBPs and ORs of *An. stephensi* and *An. gambiae* as well as *Ae. aegypti's*. No homology was found with *D. melanogaster* OBPs and ORs. As an important component of the olfactory system, correctly identifying a species' OBPs and ORs provide a valuable resource for downstream translational research that will ultimately aim to better control the malaria vector *An. stephensi*.

**Funding:** The author(s) received no specific funding for this work.

**Competing interests:** The authors have declared that no competing interests exist.

## Introduction

*Anopheles stephensi*, is vector of several *Plasmodium* species causing malaria [1,2]. Despite advances in prophylactic treatments, malaria continues to impose heavy burdens on global healthcare systems. *An. stephensi* has a wide geographic distribution throughout the South Asia, the Middle East and recently East Africa [3–7]. *An. stephensi* predominately transmits malaria in urban areas. Thus, control of *An. stephensi* can significantly reduce the malaria transmission. Olfaction mechanisms are central to insects including *An. stephensi* for host-seeking, oviposition and feeding [8,9]. Olfactory sensing mechanisms are located within the antennal sensillum of the *An. stephensi* [10,11]. Excitatory behaviour modifying compounds or semiochemicals enter sensilla through pores and cross the sensillum lymph with the aid of odorant binding proteins (OBPs) to induce olfactory sensing in insects [12].

The odorant binding proteins (OBPs) are globular in nature and specifically transport semi-ochemicals through sensillum lymph to odorant receptors (ORs) that ultimately send electrical signals to the insect brain [13,14]. OBPs also have an additional role of protecting the odorants from odorant degrading enzymes [15]. In *Antheraea polyphemus* (polyphemus moth), first OBP was identified 40 years ago [16]. Binding of semiochemicals to OBPs is pH dependent wherein binding takes place at pH6.5 whereas bound chemicals are released from OBPs near ORs where the pH measures around 4.5 [17–20].

The number of OBPs varies between insect species, with OBPs being highly expressed in sensillum lymph [21–23]. The number of OBPs identified in *Drosophila melanogaster*, *Aedes aegypti* and *Anopheles gambiae* are 51, 66 and 57, respectively [24–27]. There are several classes of OBPs: classic, atypical, plus-C, and minus-C OBPs [21]. Classification of OBPs is based on the presence of conserved cysteine residues that are considered as motifs. Six conserved cysteines are present in the classic OBPs. In each class, cysteine 2 (C2) and cysteine 3 (C3), being three amino acid residues apart, are conserved. Similarly, cysteine 5 (C5) and cysteine (C6) are conserved having eight amino acid residues between them. However, number of amino acid residues varies between C1-C2, C3-C4 and C4-C5. Classic and atypical OBPs contain same number of conserved cysteines but the number of amino acids between conserved cysteine residues vary. However, plus-C OBPs contain the two extra cysteines 4a and 6a along with a conserved proline immediately after the C6. Most of insect OBPs belong to classic group of OBPs [28]. In *An. stephensi*, two OBPs and one OR that is OR8 have been identified [29–31].

Odorant receptors (ORs) are transmembrane proteins containing seven helices like G-protein coupled receptors (GPCRs). ORs have been previously identified based on *in situ* hybridization using RNA probes and transcriptomic data from different organisms [32]. Similarly, ORs have also been identified by analysing the genomic data as in *An. gambiae* [33]. The number of ORs varies with the organisms. ORs are coupled with the Odorant Receptor co-receptors (Orco). Orco are highly conserved across all the insect species having been originally named as OR83b in *D. melanogaster* [34]. Insect olfaction relies on the interplay of OBPs that carry odorant molecules to the ORs and Orco, which results in electrical signals being sent to the insect brain.

In this study, genome wide analysis of the *An. stephensi* OBPs was performed to identify and classify the odorant binding proteins (OBPs) using the custom motifs and odorant receptors (ORs). Phylogenetic analyses were conducted to establish the evolutionary relationship of the OBPs and ORs with the closely related organisms *Ae. aegypti*, *An. gambiae*, and *D. melanogaster*. Further, we investigated synteny between the OBPs and ORs to identify the syntenic regions between these insects. This study has analyzed the gene structure and predicted the physicochemical properties and subcellular localization of the OBPs and ORs as well for confirmation of the OBPs and ORs. Being a vital component of insect olfaction, OBPs and ORs of

*An*. *stephensi* play crucial roles in the mosquito life cycle and therefore, are likely to have a major influence on the potential for the mosquito to transmit malarial diseases. This study serves as the basis for the structural and functional characterization of the OBPs and ORs of *An. stephensi*. Identified OBPs and ORs will help in understanding the pathways sensitive to attractants and repellents in the olfactory mechanism and vector control strategies. It provides the foundation for comparative studies based on olfactory mechanisms in other insect species as well.

## Methodology

The methodology is summarized as a flow chart shown in Fig 1.

### Identification of the OBPs and ORs

Protein FASTA file of the *Anopheles stephensi* Indian strain, sequenced by UC Irvine, was downloaded from the NCBI database (Refseq: GCF_013141755.1). Retrieved protein FASTA file was used as a protein database in BLAST+ software for local BLASTp using the OBP sequences of *Ae. aegypti*, *An. gambiae* and *D. melanogaster* as query sequences. Custom motifs were designed on earlier studies according to the ScanProsite input format. These motifs are based on the conserved cysteine residues present in three classes of OBPs: classic [Cx(15,39)Cx(3)Cx(21,44)Cx(7,12)Cx(8)C], atypical [Cx(26,27)Cx(3)Cx(36,38)Cx(11,15)Cx(8)C], and plus-C [Cx(8,41)Cx(3)Cx(39,47)Cx(17,29)Cx(9)Cx(8)CPx(9,11)C]. BLASTp output was used to detect the presence of these motifs using ScanProsite (https://prosite.expasy.org/scanprosite/) to extract OBPs sequences based on their classification [35]. The retrieved sequences were

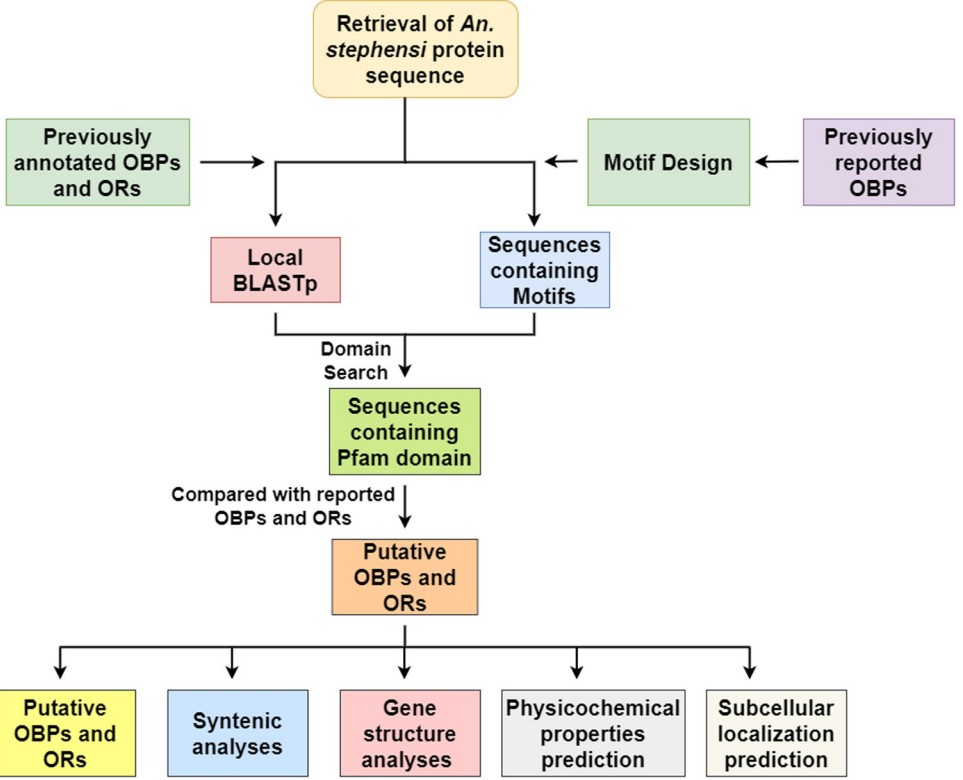

**Fig 1. Pipeline of the methodology.** Pipeline of the methodology used for this study is represented.

further analyzed to detect the presence of the PBP/GOBP domain (Pfam: PF01395) using Pfam web server (http://pfam.xfam.org/). Pfam is a large collection of the protein families and domains [36]. Retrieved OBPs sequences were used to perform a BLASTp search against protein database of *An. stephensi* to search for the missing OBPs sequences in. OBPs were named according to their position on the chromosomes as AsteOBP. Positions of the OBP encoding genes on *An. stephensi* chromosomes were visualized using phenogram tool (http://visualization.ritchielab.org/phenograms/plot) [37]. Whereas open reading frame (ORF) were predicted using the ORFfinder (https://www.ncbi.nlm.nih.gov/orffinder/).

Like OBPs, OR sequences of the *Ae. aegypti*, *An. gambiae* and *D. melanogaster* were used as the query sequences for the local BLASTp [38]. The protein FASTA file of *An. stephensi* was used as the database. BLASTp output sequences were further checked for the presence of the 7tm_6 domain (Pfam ID: PF02949) using the Pfam web server. Further, redundant sequences having 100% sequence identity were removed. Similarly, positions of OR encoding genes on *An. stephensi* chromosomes were visualized using phenogram.

## Multiple sequence alignment and phylogenetic tree analyses

Multiple sequence alignment for the OBPs was done using Clustal-W in Mega X-V10.2 with gap opening penalty 10. The gap extension penalty was set to 0.1 for pairwise alignment and 0.2 for the multiple sequence alignment [39]. Multiple sequence alignment was performed separately for the classic, atypical and plus-C OBPs to visualize the motif regions in each OBP subtype. Multiple sequence alignments were visualized using Jalview [40].

Multiple sequence alignments of the OBPs of the *An. stephensi*, *An. gambiae*, *Ae. aegypti* and *D. melanogaster* were performed using the Clustal-W on the Galaxy server (https://usegalaxy.org/). This alignment was used for the generation of the Maximum-Likelihood tree using the FastTree 2 on the galaxy webserver [41]. Jones-Taylor-Thornton 1992 model (JTT-Model) was used as evolutionary model that uses protein sequences for the faster generation of mutation metrices of proteins. The tree was further modified using the iTOL web server [42]. Similarly, Clustal W and FastTree 2 were used for the construction of phylogenetic tree of the Odorant Receptors (ORs).

## Synteny prediction of the OBPs and ORs

Synteny analysis was performed using TBTools [43]. TBTools uses MCScanX to find the syntenic regions between the chromosomes of two organisms [44]. MCScanX is an algorithm that is used to scan the multiple genomes and identify putative homologous chromosomal regions by aligning those using genes as anchors. OBPs of the *An. stephensi* were analyzed against the genome of *An. gambiae* (Genbank: GCA_000005575.1), *Ae. aegypti* (Genbank: GCA_002204515.1) and *D. melanogaster* (Genbank: GCA_000001215.4) to identify the collinear blocks between their genomes. Likewise, syntenic regions were also predicted in the ORs of *An. stephensi* with the ORs of *An. gambiae*, *Ae. aegypti* and *D. melanogaster*.

## Gene structure analysis of OBPs and ORs

Gene structures of the OBPs and ORs were visualized using the TBTools [43]. A Genome Feature File (GFF) file of the genomic sequences of *An. stephensi* was used for the identification of the Untranslated Regions (UTRs) and Coding DNA Sequences (CDS) in OBPs and ORs genes. MEME web server (https://meme-suite.org/meme/tools/meme) was used for the presence of conserved motifs in the peptides sequences of the OBPs and ORs [45]. Conserved Domain Database (CDD) was used to visualize the conserved Pfam domains in peptide sequences of OBPs and ORs [46]. Phylogenetic tree of OBPs and ORs of *An. stephensi* was

constructed using the Neighbor Joining Method in Mega X-V10.2 for the use of cladogram in the gene structure representation.

### Physiochemical properties and sub-cellular localization prediction of OBPs and ORs

Physiochemical properties of the OBPs and ORs including molecular weight and isoelectric points were predicted using ProtParam in the Expasy webserver (https://web.expasy.org/protparam/). Sub-cellular localization of the OBPs and ORs was predicted using the WoLF PSORT (https://wolfpsort.hgc.jp/) and CELLO web server (http://cello.life.nctu.edu.tw/cgi/main.cgi).

## Results

### Identification of OBPs and ORs

A total 44 OBPs were identified in the *An. stephensi* having complete conserved motifs. There were 27 classic, 10 atypical, and 7 plus-C OBPs as their NCBI peptide accessions and gene IDs are represented in the Table 1. Genomic and protein sequences of putative OBPs are given in S1 and S2 Data, respectively. Similarly, ORF length and number of amino acids in the OBPs have been provided in the S1 Table.

Further, putative ORs identified in the genome of the *An. stephensi* were 45. These genes were renamed according to their respective position on the chromosomes of *An. stephensi.* The name, NCBI protein accessions and gene IDs of the sequences are presented in the Table 2. Genomic and peptide sequences of putative ORs are given in S3 and S4 Data, respectively. Similarly, ORF length and number of amino acids in the ORs have been provided in the S2 Table.

### Chromosomal location of the OBPs and ORs

Chromosomal location of the OBPs was depicted using the Phenogram web server as shown in the Fig 2. The highest numbers of the OBPs were present on chromosome 2 containing 26 OBPs, whereas chromosome X and chromosome 3 contained 6 and 11 OBPs, respectively. Eight OBPs, starting from AsteOBP7 to AsteOBP14, were clustered on chromosome 2. Whilst on chromosome X, AsteOBP3, AsteOBP4, AsteOBP5 and AsteOBP6 were clustered together. Similarly, AsteOBP33-AsteOBP35 and AsteOBP36-AsteOBP38 were clustered on the chromosome 3. Chromosome X only contained the classic OBPs whereas chromosome 2 and 3 contained all three types. AsteOBP44 gene had not been placed on the chromosomes to date due to limitations of genomic assembly.

ORs were clustered on chromosome 2 of *An. stephensi* as there were 21 ORs' gene located on it as represented in Fig 3. Chromosome X contained the least OR gene as it contained only 4 genes. Chromosome 3 contained 19 OR genes. Most of the OR genes were scattered except a few that were clustered together. AsteOR2-AsteOR4 were clustered together on the chromosome X. Similarly, some genes were clustered in pairs at the chromosome 2. But on chromosome 3, only three clusters of the OR genes were present where each cluster had three OR genes. AsteOR45 was present on a scaffold that was not placed on the chromosomes the genome assembly.

### Multiple sequence alignment and phylogenetic analysis of the OBPs and ORs

Alignments are presented in Fig 4. Clear motifs patterns depicting the conserved cysteines were visible in all three types of OBPs. Phylogenetic tree was constructed using the maximum-

**Table 1. Classification and NCBI accessions of identified OBPs.**

| OBP | Protein Accession | Gene ID | Classes of OBPs | Chromosome | Molecular weight | Isoelectric Point |
| --- | --- | --- | --- | --- | --- | --- |
| AsteOBP1 | XP_035917841.1 | LOC118504557 | Classic | X | 18669 | 4.7 |
| AsteOBP2 | XP_035891203.1 | LOC118504570 | Classic | X | 16593 | 6.5 |
| AsteOBP3 | XP_035891204.1 | LOC118510310 | Classic | X | 16564 | 5.6 |
| AsteOBP4 | XP_035891207.1 | LOC118510318 | Classic | X | 16813 | 5.6 |
| AsteOBP5 | XP_035891740.1 | LOC118510325 | Classic | X | 15898 | 4.2 |
| AsteOBP6 | XP_035891741.1 | LOC118510331 | Classic | X | 14940 | 6.2 |
| AsteOBP7 | XP_035892015.1 | LOC118502745 | Classic | 2 | 15105 | 8.4 |
| AsteOBP8 | XP_035892546.1 | LOC118502746 | Atypical | 2 | 50208 | 8.2 |
| AsteOBP9 | XP_035892547.1 | LOC118502748 | Atypical | 2 | 39006 | 5.1 |
| AsteOBP10 | XP_035893080.1 | LOC118503023 | Classic | 2 | 17788 | 4.9 |
| AsteOBP11 | XP_035894183.1 | LOC118503024 | Plus-C | 2 | 22832 | 4.7 |
| AsteOBP12 | XP_035894335.1 | LOC118503161 | Classic | 2 | 15761 | 4.3 |
| AsteOBP13 | XP_035894381.1 | LOC118503418 | Classic | 2 | 16437 | 6.5 |
| AsteOBP14 | XP_035895118.1 | LOC118503419 | Atypical | 2 | 31279 | 8.4 |
| AsteOBP15 | XP_035895684.1 | LOC118503662 | Classic | 2 | 15844 | 6.3 |
| AsteOBP16 | XP_035895818.1 | LOC118504157 | Classic | 2 | 17738 | 8.9 |
| AsteOBP17 | XP_035895821.1 | LOC118504227 | Classic | 2 | 15028 | 4.8 |
| AsteOBP18 | XP_035897326.1 | LOC118504259 | Atypical | 2 | 24233 | 6.6 |
| AsteOBP19 | XP_035897725.1 | LOC118504578 | Classic | 2 | 16412 | 5.2 |
| AsteOBP20 | XP_035898485.1 | LOC118504803 | Classic | 2 | 16227 | 8.1 |
| AsteOBP21 | XP_035898834.1 | LOC118504874 | Classic | 2 | 19676 | 8.9 |
| AsteOBP22 | XP_035898836.1 | LOC118504876 | Classic | 2 | 16709 | 8.7 |
| AsteOBP23 | XP_035899548.1 | LOC118505522 | Classic | 2 | 17444 | 5.8 |
| AsteOBP24 | XP_035900970.1 | LOC118505693 | Classic | 2 | 13982 | 8.4 |
| AsteOBP25 | XP_035902367.1 | LOC118506007 | Classic | 2 | 16094 | 4.8 |
| AsteOBP26 | XP_035903187.1 | LOC118506171 | Classic | 2 | 14920 | 4.7 |
| AsteOBP27 | XP_035903789.1 | LOC118506172 | Classic | 2 | 18610 | 4.6 |
| AsteOBP28 | XP_035905778.1 | LOC118506477 | Classic | 2 | 17744 | 9.3 |
| AsteOBP29 | XP_035907589.1 | LOC118507089 | Plus-C | 2 | 21790 | 8.5 |
| AsteOBP30 | XP_035907591.1 | LOC118507680 | Plus-C | 2 | 21161 | 5.9 |
| AsteOBP31 | XP_035907600.1 | LOC118507965 | Plus-C | 2 | 22467 | 8.8 |
| AsteOBP32 | XP_035908250.1 | LOC118508265 | Plus-C | 2 | 23618 | 7.5 |
| AsteOBP33 | XP_035908252.1 | LOC118509362 | Plus-C | 3 | 19688 | 4.9 |
| AsteOBP34 | XP_035908253.1 | LOC118510176 | Plus-C | 3 | 19627 | 4.9 |
| AsteOBP35 | XP_035909003.1 | LOC118510178 | Atypical | 3 | 36926 | 6 |
| AsteOBP36 | XP_035910841.1 | LOC118510185 | Classic | 3 | 17618 | 8.4 |
| AsteOBP37 | XP_035914753.1 | LOC118510478 | Atypical | 3 | 31775 | 5.3 |
| AsteOBP38 | XP_035917598.1 | LOC118510479 | Classic | 3 | 15299 | 4.8 |
| AsteOBP39 | XP_035895071.1 | LOC118510480 | Classic | 3 | 20736 | 5.2 |
| AsteOBP40 | XP_035895083.1 | LOC118510798 | Classic | 3 | 15612 | 5.3 |
| AsteOBP41 | XP_035907843.1 | LOC118511635 | Atypical | 3 | 35623 | 5.8 |
| AsteOBP42 | XP_035907857.1 | LOC118513288 | Atypical | 3 | 33435 | 5.2 |
| AsteOBP43 | XP_035907868.1 | LOC118514663 | Atypical | 3 | 33188 | 5.2 |
| AsteOBP44 | XP_035907882.1 | LOC118515263 | Atypical | Unknown | 31914 | 5.4 |

Classification of the identified odorant binding proteins (OBPs) along with their NCBI peptide accession and gene IDs are presented in the table. Molecular weight and isoelectric point of OBPs are also given.

**Table 2. Renaming of odorant receptor (ORs) and chromosome.**

| ORs | NCBI Protein Accession | NCBI Gene ID | Chromosome | Molecular weight | Isoelectric Point |
|---|---|---|---|---|---|
| AsteOR1 | XP_035892088.1 | LOC118502993 | 1 | 47.484 | 9.49 |
| AsteOR2 | XP_035917717.1 | LOC118514681 | 1 | 47.46 | 9.51 |
| AsteOR3 | XP_035900263.1 | LOC118506771 | 1 | 11.761 | 9.51 |
| AsteOR4 | XP_035918152.1 | LOC118516006 | 1 | 29.498 | 8.69 |
| AsteOR5 | XP_035893454.1 | LOC118503850 | 2 | 50.216 | 9.17 |
| AsteOR6 | XP_035893486.1 | LOC118503863 | 2 | 30.798 | 7 |
| AsteOR7 | XP_035901553.1 | LOC118507296 | 2 | 45.165 | 8.73 |
| AsteOR8 | XP_035899159.1 | LOC118506317 | 2 | 97.083 | 9.08 |
| AsteOR9 | XP_035898901.1 | LOC118506200 | 2 | 49.355 | 8.23 |
| AsteOR10 | XP_035898518.1 | LOC118506022 | 2 | 45.771 | 8.7 |
| AsteOR11 | XP_035901800.1 | LOC118507433 | 2 | 46.713 | 6.22 |
| AsteOR12 | XP_035903051.1 | LOC118507903 | 2 | 46.521 | 6.55 |
| AsteOR13 | XP_035891168.1 | LOC118502720 | 2 | 46.096 | 5.9 |
| AsteOR14 | XP_035899865.1 | LOC118506605 | 2 | 52.647 | 8.83 |
| AsteOR15 | XP_035893072.1 | LOC118503656 | 2 | 39.869 | 9.08 |
| AsteOR16 | XP_035893066.1 | LOC118503653 | 2 | 47.581 | 9.26 |
| AsteOR17 | XP_035895309.1 | LOC118504650 | 2 | 14.085 | 6.7 |
| AsteOR18 | XP_035890596.1 | LOC118502471 | 2 | 43.074 | 7.17 |
| AsteOR19 | XP_035890597.1 | LOC118502472 | 2 | 48.388 | 8.52 |
| AsteOR20 | XP_035897380.1 | LOC118505546 | 2 | 43.686 | 7.6 |
| AsteOR21 | XP_035890600.1 | LOC118502475 | 2 | 51.037 | 9.41 |
| AsteOR22 | XP_035903785.1 | LOC118508262 | 2 | 53.873 | 8.11 |
| AsteOR23 | XP_035903780.1 | LOC118508261 | 2 | 55.875 | 8.31 |
| AsteOR24 | XP_035890608.1 | LOC118502484 | 2 | 46.469 | 8.91 |
| AsteOR25 | XP_035896260.1 | LOC118505078 | 2 | 46.133 | 7.55 |
| AsteOR26 | XP_035914390.1 | LOC118513096 | 3 | 46.464 | 9.36 |
| AsteOR27 | XP_035913964.1 | LOC118512920 | 3 | 48.938 | 6.37 |
| AsteOR28 | XP_035910189.1 | LOC118511343 | 3 | 53.323 | 6.59 |
| AsteOR29 | XP_035904929.1 | LOC118509018 | 3 | 42.07 | 8.72 |
| AsteOR30 | XP_035904961.1 | LOC118509031 | 3 | 43.592 | 9.29 |
| AsteOR31 | XP_035904965.1 | LOC118509034 | 3 | 43.903 | 5.75 |
| AsteOR32 | XP_035907269.1 | LOC118509993 | 3 | 44.707 | 8.63 |
| AsteOR33 | XP_035907274.1 | LOC118509997 | 3 | 17.289 | 4.97 |
| AsteOR34 | XP_035907276.1 | LOC118509999 | 3 | 43.583 | 5.65 |
| AsteOR35 | XP_035904780.1 | LOC118508959 | 3 | 10.478 | 8.73 |
| AsteOR36 | XP_035907277.1 | LOC118510000 | 3 | 32.545 | 6.1 |
| AsteOR37 | XP_035910820.1 | LOC118511624 | 3 | 48.887 | 8.47 |
| AsteOR38 | XP_035908757.1 | LOC118510691 | 3 | 43.658 | 8.99 |
| AsteOR39 | XP_035908756.1 | LOC118510690 | 3 | 43.438 | 8.56 |
| AsteOR40 | XP_035908759.1 | LOC118510692 | 3 | 43.242 | 9.03 |
| AsteOR41 | XP_035908990.1 | LOC118510790 | 3 | 15.858 | 5.27 |
| AsteOR42 | XP_035907332.1 | LOC118510050 | 3 | 32.201 | 8.45 |
| AsteOR43 | XP_035907338.1 | LOC118510056 | 3 | 45.4 | 9.52 |
| AsteOR44 | XP_035916306.1 | LOC118513983 | 3 | 33.276 | 5.81 |
| AsteOR45 | XP_035919246.1 | LOC118517353 | Unknown | 31.278 | 6.55 |

Identified odorant receptors (ORs) along with their NCBI peptide accession, chromosome and gene IDs are presented in the table. Molecular weight and isoelectric points of ORs are also given.

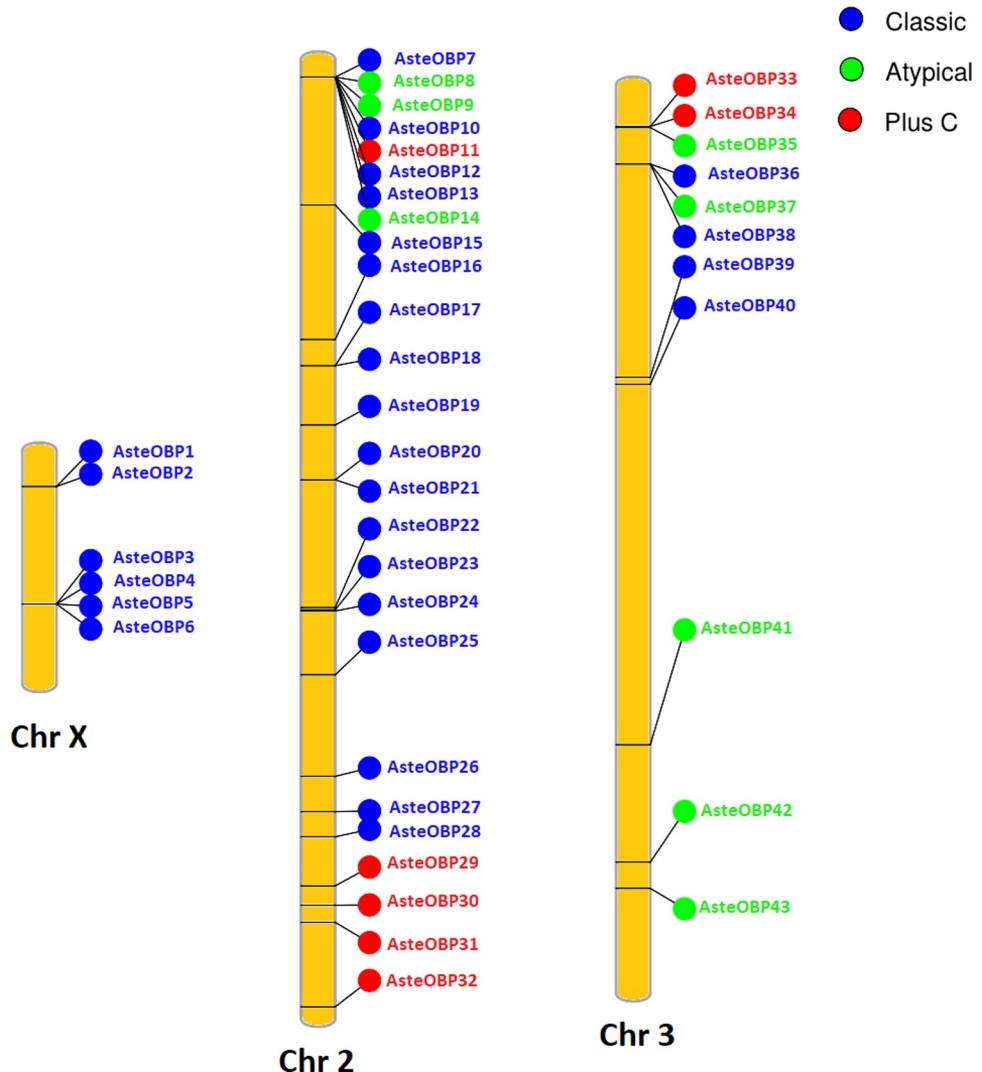

**Fig 2. Chromosomal location of Odorant Binding Proteins (OBPs).** Aste**OBPs are shown based on their position on chromosomes of the *An. stephensi*. Chromosome 2 contained highest number of OBP genes. Some OBPs are clustered on the chromosomes.**

likelihood method. OBP sequences of the *An. stephensi*, *An. gambiae*, *D. melanogaster*, and *Ae. aegypti* were used for the construction of phylogenetic tree in Mega-X as shown in Fig 5. Most of the OBPs showed close relationship with OBPs of *An. gambiae* but, none of the OBPs showed close relationship with the OBPs of *D. melanogaster*. Similarly, only AsteOBP1, AsteOBP5, AsteOBP21, AsteOBP27, and AsteOBP31 were closely related to the AaegOBP20, AaegOBP15, AaegOBP4, AaegOBP59 and AaegOBP5 of the *Ae. aegypti*, respectively. It confirms the close relationship of the *An. stephensi* with the *An. gambiae* whereas *D. melanogaster* was distantly rooted in the phylogenetic tree among the compared organisms.

ORs showed close phylogenetic relationship with the ORs of *An. gambiae* as represented in Fig 6. However, only two AsteORs: AaegOR28 and AaegOR23 were closely related to AaegORs: AaegOR5 and AaegOR7, respectively. None of the *An. stephensi* ORs were closely rooted to the *D. melanogaster* ORs.

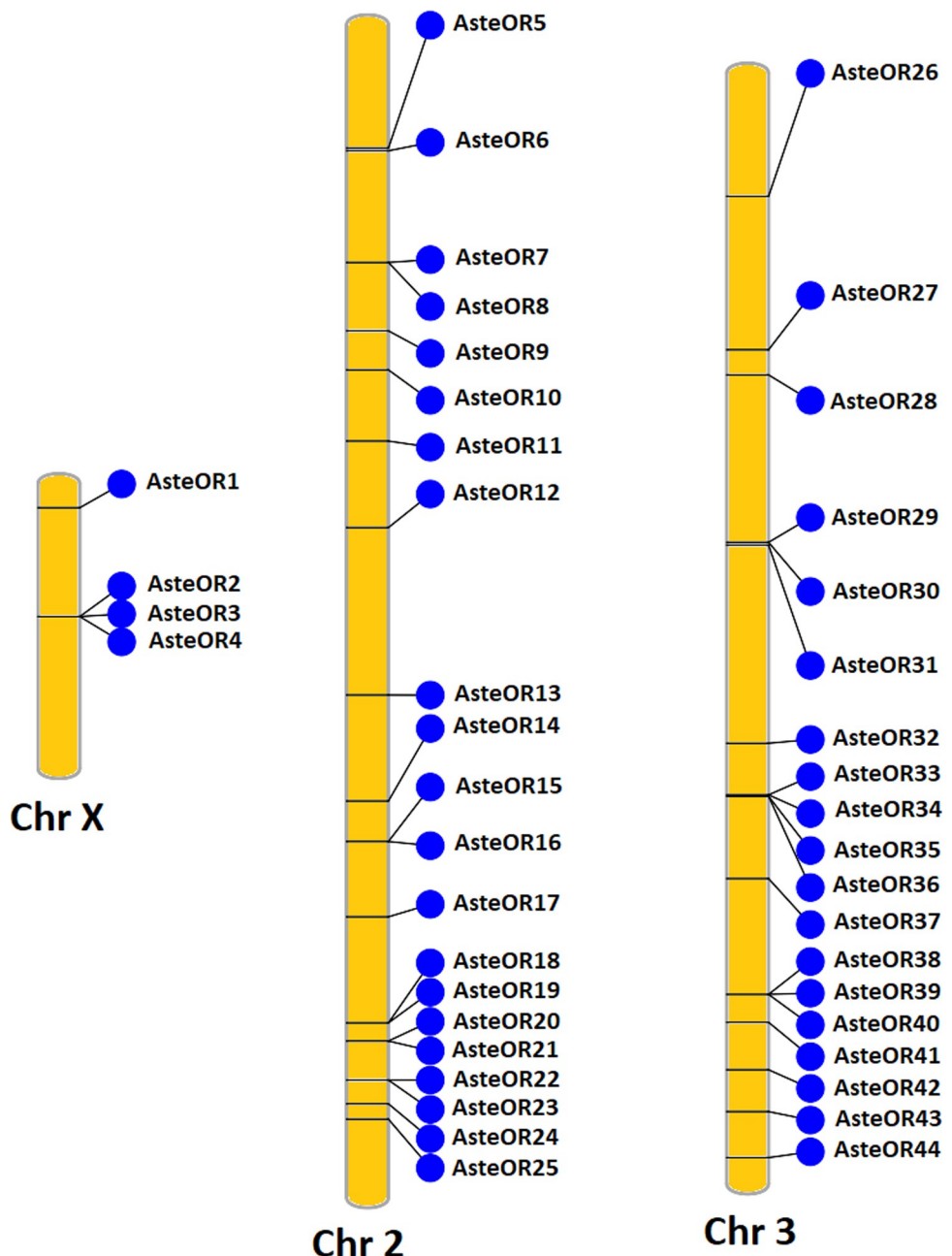

**Fig 3. Chromosomal location of Odorant Receptors (ORs).** Position of the OR genes is shown on the chromosomes of the *An. stephensi*. Chromosome 2 contained highest number of OR genes. Some ORs are clustered on the chromosomes.

## Synteny analysis of OBPs and ORs

Synteny analysis was performed between the *An. stephensi* and *Ae. aegypti*. Total 11 OBPs of *An. stephensi* had the syntenic region with the OBPs of the *Ae. aegypti* as shown in the Fig 7A. OBPs present on the chromosome X of the *An. stephensi* and *Ae. aegypti* did not share any orthologues. Syntenic regions were present between the OBPs of the chromosome 2 of *An.*

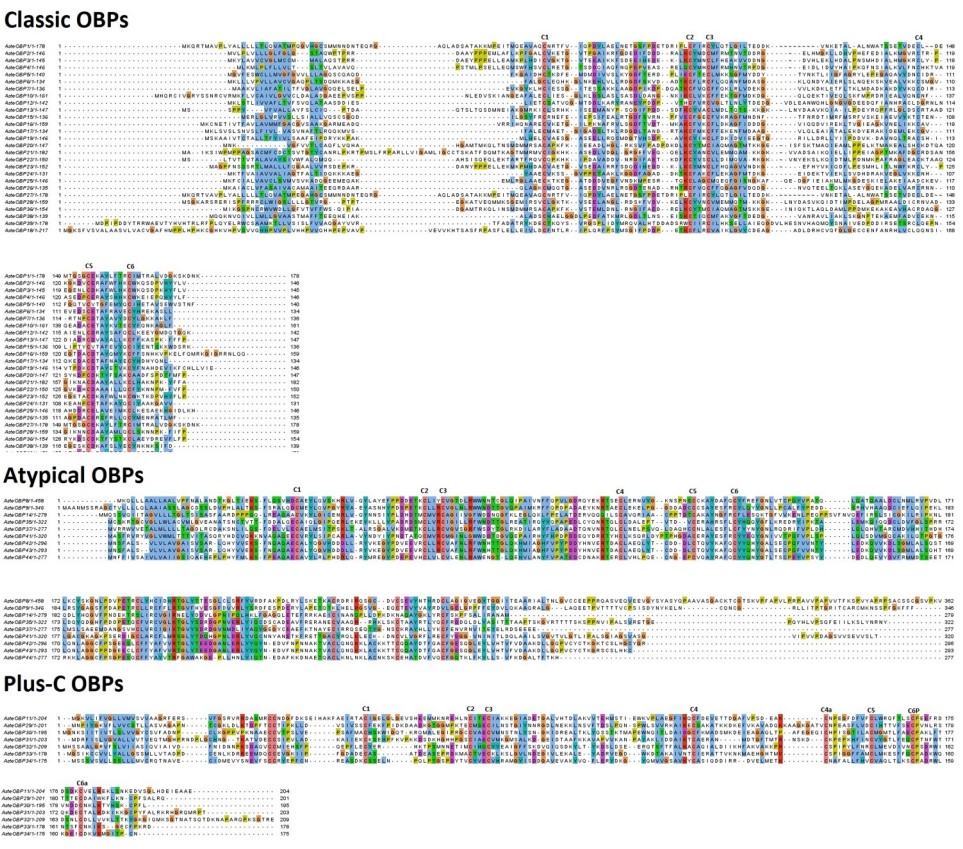

**Fig 4. Multiple sequence alignment of the OBPs.** Multiple Sequence Alignment of the OBPs is shown in figure. Conserved motifs have been highlighted in all three classes of the OBPs: Classic, atypical and plus-C OBPs.

*stephensi* and the OBPs present on the chromosome 3 of the *Ae. aegypti* whereas chromosome 3 OBPs of *An. stephensi* contained collinear with the OBPs of chromosome 2 of *Ae. aegypti*.

Synteny analysis were also performed between the *An. stephensi* and *An. gambiae* genome as represented in Fig 7B. A total of 29 OBPs of the *An. stephensi* contained the collinear blocks with the OBPs of *An. gambiae*. Four syntenic regions were present between the chromosome X of both organisms. Whereas others were distributed between chromosome 2 and chromosome 3 of the *An. stephensi* and chromosome 2L, 2R, 3L and 3R of the *An. gambiae*. In comparison to *An. gambiae* OBPs, OBPs of *Ae. aegypti* had less synteny with the OBPs of *An. stephensi*.

Syntenic analyses of the ORs of *An. stephensi* and *Ae. aegypti* are shown in Fig 8A. A total of 12 ORs of *An. stephensi* shared syntenic regions with 13 OR genes of the *Ae. aegypti*. Only one OR2 gene present on the chromosome X of *An. stephensi* showed homologous region with the OR41 of the *Ae. aegypti*. OR genes present on the chromosome 2 of *An. stephensi* showed close homology with the ORs present on the Chromosome X and 3 of *Ae. aegypti*. However, OR15 on chromosome 2 shared homology with the two ORs: OR8 and OR37 of *Ae. aegypti*. Whereas OR genes present on the chromosome 3 of *An. stephensi* were homologous to ORs of chromosome 2 of *Ae. aegypti*.

Synteny analysis of *An. stephensi* with *An. gambiae* ORs is shown in Fig 8B. A total of 32 ORs of *An. stephensi* shared homology with 34 ORs of *An. gambiae*. OR1 gene present on the chromosome X of *An. stephensi* showed homology with the OR36 and OR52 genes present on the chromosome X of *An. gambiae*. However, ORs present on the chromosome 2 of the *An. stephensi* were homologous to the ORs present on chromosome 2R and 3L of *An. gambiae*. But

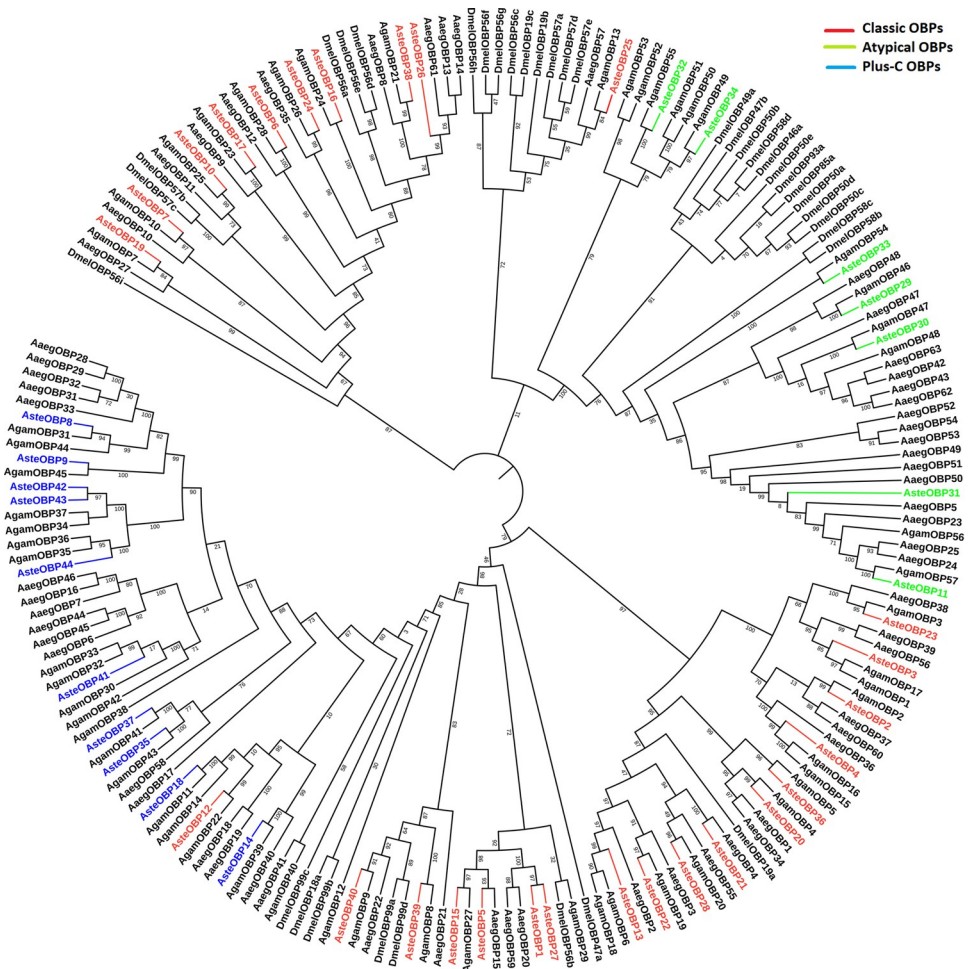

**Fig 5. Phylogenetic analysis of OBPs.** Phylogenetic analysis of the OBPs of *Anopheles stephensi*, *Anopheles gambiae*, *Aedes aegypti* and *Drosophila melanogaster* were carried out using FastTree 2. *An. stephensi* OBPs showed closer relationship to *An. gambiae* OBPs. The bootstrap values have been represented in the figure.

ORs present on the chromosome 3 of the *An. stephensi* shared syntenic region with the ORs of chromosome 2L and 3R in *An. gambiae*. None of the *An. stephensi* OBPs and ORs shared homology with the OBPs and ORs of *D. melanogaster*.

## Gene structure analysis of OBPs and ORs

Gene structure analysis of the OBPs was carried out using the TBTool as depicted in the Fig 9. Conserved motifs present in the OBPs were identified using the MEME web server. Motif 1 was present in all the OBPs. Whereas motifs 2, 3 and 4 were present in the most of the OBPs. Conserved domains were identified using the Conserved Domain Database (CDD). CDD results indicated OBPs 11, 30, 31, 32, 33 and 34 had insignificant similarity with the Pfam PBP/GOBP domain but these were Plus-C OBPs having complete motifs. Whereas all other OBPs contained the PBP/GOBP domains.

Gene structure of the ORs of *An. stephensi* has been represented in the Fig 10. 7tm_6 domain was present in all the sequences. CDS were present in the sequences whereas some sequences lacked UTRs. Motif 2 and Motif 3, predicted by MEME, were present in most of the sequences except OR36. Whereas motif 1, 4, and 5 were present in few sequences.

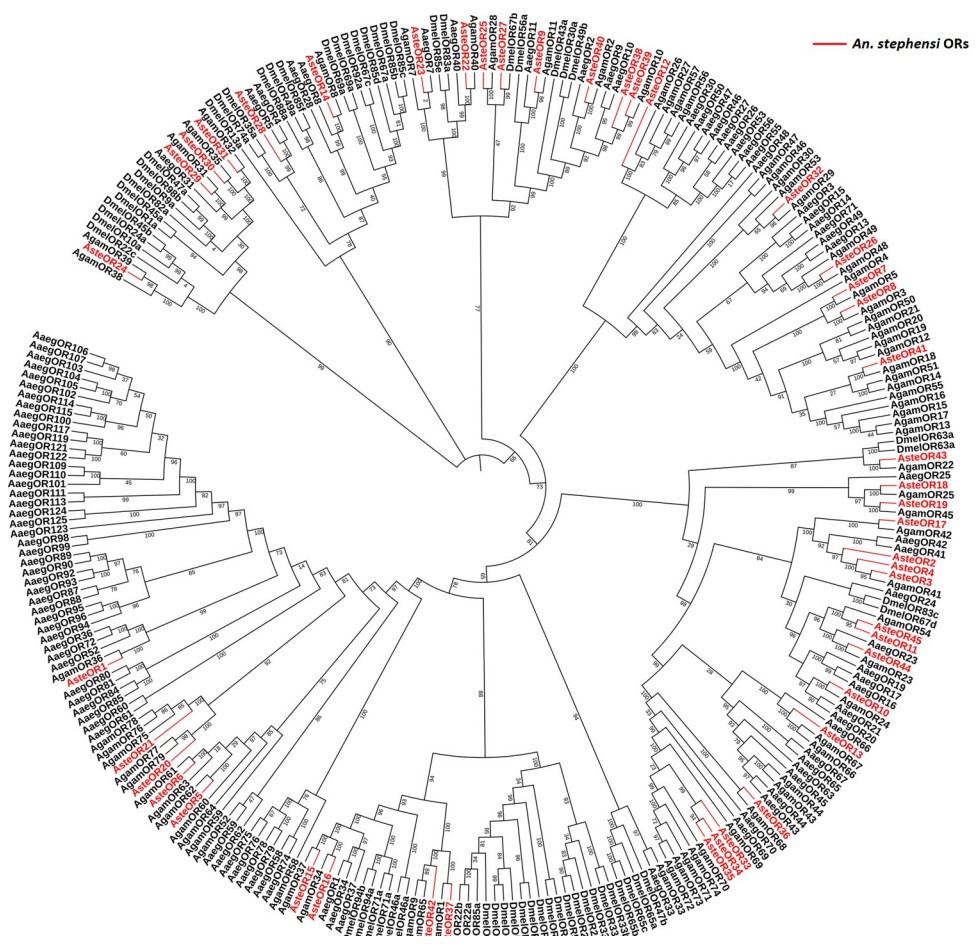

**Fig 6. Phylogenetic analysis of ORs.** Phylogenetic analysis of *Anopheles stephensi*, *Anopheles gambiae*, *Aedes aegypti* and *Drosophila melanogaster* ORs were carried out using FastTree 2 on galaxy web server. *An. stephensi* ORs showed closer relationship to *An. gambiae* ORs. The bootstrap values have been represented in the figure.

## Prediction of physicochemical properties and subcellular localization of the OBPs and ORs

The molecular weight of the atypical OBPs varied from the 24.2 kDa to 39.0 kDa except for the OBP8 that had 50.2 kDa as presented in Table 1. In classic OBPs, molecular weight ranged between the 14 kDa to 20.7 kDa. Plus-C OBPs had the molecular weight in the range of 19.7–23.6 kDa. Isoelectric point, a pH at which molecule is neutral, ranged from 4.2 to 8.9 for all the OBPs as given in Table 1.

The molecular weight of the ORs varied from 30.798 kDa to 97.083 kDa for OR6 and OR8 respectively as shown in Table 2. However, some sequences that had partial 7tm_6 (Odorant Receptor Domain) had less molecular weight. Isoelectric point ranged between the 4.97 and 9.52 for OR33 and OR43, respectively. Isoelectric points of the OBPs and ORs in relation to the molecular weight are shown in the Fig 11.

CELLO and WoLF PSORT predicted extra-cellular localization of the OBPs indicating that OBPs are secretory proteins whereas ORs were predicted to be membrane embedded that proved their transmembrane propensity.

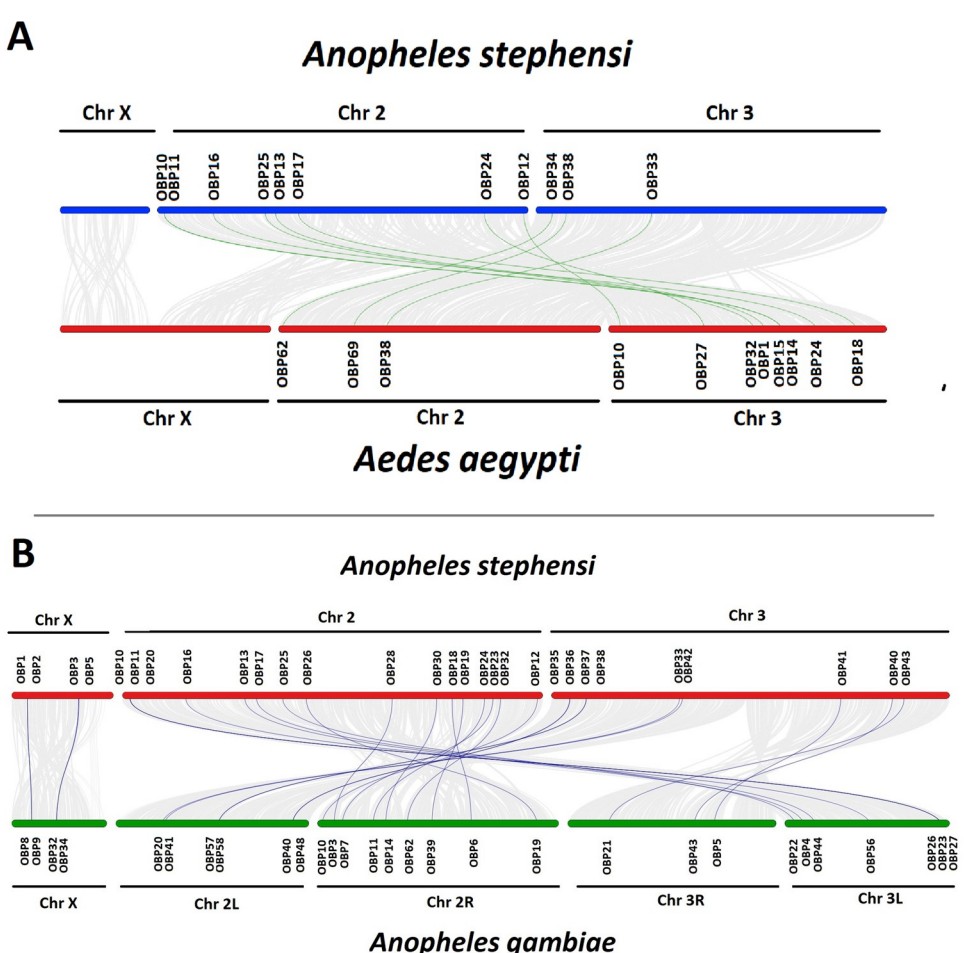

**Fig 7. Synteny analysis of OBPs.** Synteny analysis were carried out to find the collinear blocks between the OBPs of *An. stephensi* and OBPs of (A) *Ae. aegypti* and (B) *Ae. gambiae*. Chromosomes and chromosomal position of the OBPs have been shown. Green lines (A) show the syntenic relationship between the OBPs of *An. stephensi* and *Ae. aegypti* whereas blue lines (B) represent the syntenic relationship between the OBPs *of An. stephensi* and *An. gambiae*.

## Discussion

In *An. stephensi*, 44 OBPs are identified. There are 27 classic, 10 atypical and 7 plus-C OBPs. Similarly, a total of 66 OBP encoding genes were identified in the *Ae. aegypti*, 66 in *An. gambiae*, and 49 in *D. melanogaster* [47–49]. Like *An. stephensi*, classic OBPs are abundant than atypical and plus-C OBPs in other insect species including *Ae. aegypti* and *An. gambiae* as well [14,50,51]. There were total 41, 29, and 30 classic OBPs identified in *Ae. aegypti*, *An. gambiae*, and *D. melanogaster* respectively [25,26,52]. Atypical OBPs are more abundant than the plus-C OBPs in insects. In *Ae. aegypti*, there were more plus-C OBPs than the atypical contradicting with other insects. There were 10, 16 and 12 atypical OBPs identified in *Ae. aegypti*, *An. gambiae*, and *D. melanogaster*, respectively. However, plus-C OBPs were 16, 12 and 12 in *Ae. aegypti*, *An. gambiae*, and *D. melanogaster*, respectively. Odorant Receptors (ORs) also vary in their number between insect species. A total of 45 ORs have been identified in the *An. stephensi*. Like *An. stephensi*, the numbers of identified ORs were 75 in *An. gambiae*, 61 in *D. melanogaster*, and 131 in *Ae. aegypti* [33,53,54]. However, a total of 226 ORs were identified in *Aenasius bambawalei* belonging to order Hymenoptera [55]. Some species can contain less

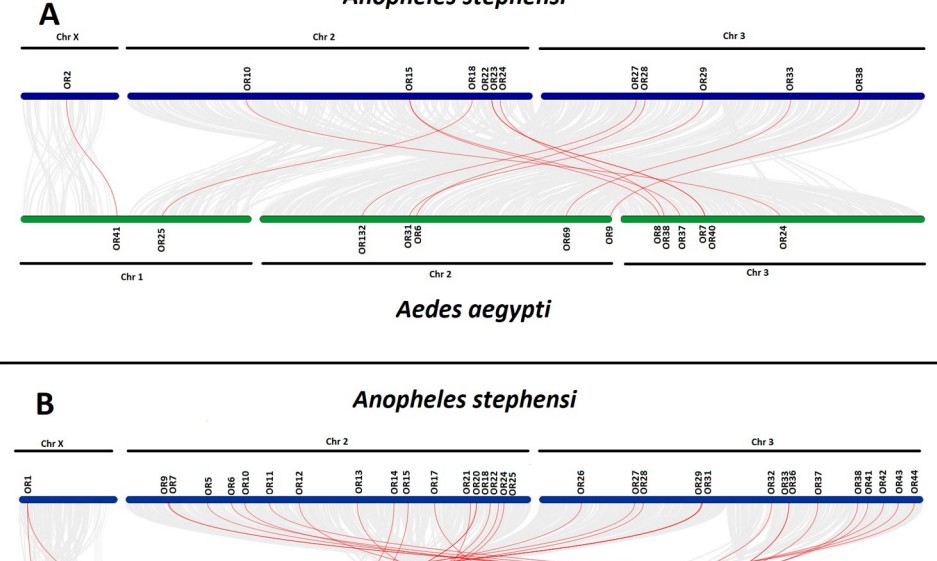

**Fig 8. Synteny analysis of ORs.** Synteny analysis were carried out to find the collinear blocks between the ORs of *An. stephensi* and ORs of (A) *Ae. aegypti* and (B) *Ae. gambiae*. Chromosomes and chromosomal position of the ORs have been shown along with the lines showing synteny between the ORs. Red lines represent the syntenic relationship between the ORs of *An. stephensi* with *Ae. aegypti* (A) and *An. gambiae* (B).

ORs, for instance, only 8 ORs have been identified in *Tomicus yunnanensis* that belongs to order Coleoptera [56].

Most of the OBP genes are localized on chromosome 2 of *An. stephensi*. Whereas Chromosome X contains six OBP genes. Similarly, the least number of OBP genes were found on chromosome X whereas most were present on chromosome 3L in *An. gambiae* [52]. Clusters of OBP encoding genes are present on all chromosomes whereas some of these genes are scattered all over the chromosomes in *An. stephensi*. A similar pattern of clustering has been reported for *D. melanogaster* that have four clusters of OBPs genes [25]. Like OBPs, ORs are also abundant on chromosome 2 whereas chromosome X contains only 4 OR genes in *An. stephensi*. This indicates a pattern that odorant genes are abundant on the chromosome 2 whereas chromosome X contains only few odorant genes in the insects.

Phylogenetic analysis showed a close relationship of the OBPs and ORs of *An. stephensi* with *An. gambiae* as both are closely related organisms. Whereas few OBPs and ORs of *An. stephensi* showed relationship with the OBPs and ORs of *Ae. aegypti*. However, OBPs and ORs were distantly related to the OBPs and ORs of the *D. melanogaster* among compared organisms. Because *An. stephensi* shares same genus with *An. gambiae* whereas *Ae. aegypti* belong to different genus. However, *D. melanogaster* belonged to different family as compared to *An. stephensi*. Like *Anopheles stephensi*, OBPs and ORs of the *Ae. aegypti* also showed close relationship with *An. gambiae* OBPs and ORs because of the closer relationship between both organism than the distantly related *D. melanogaster* [26]. Likewise, *Culex quinquefasciatus* OBPs and ORs showed the closer relationship with OBPs and ORs of *Ae. aegypti* as compared

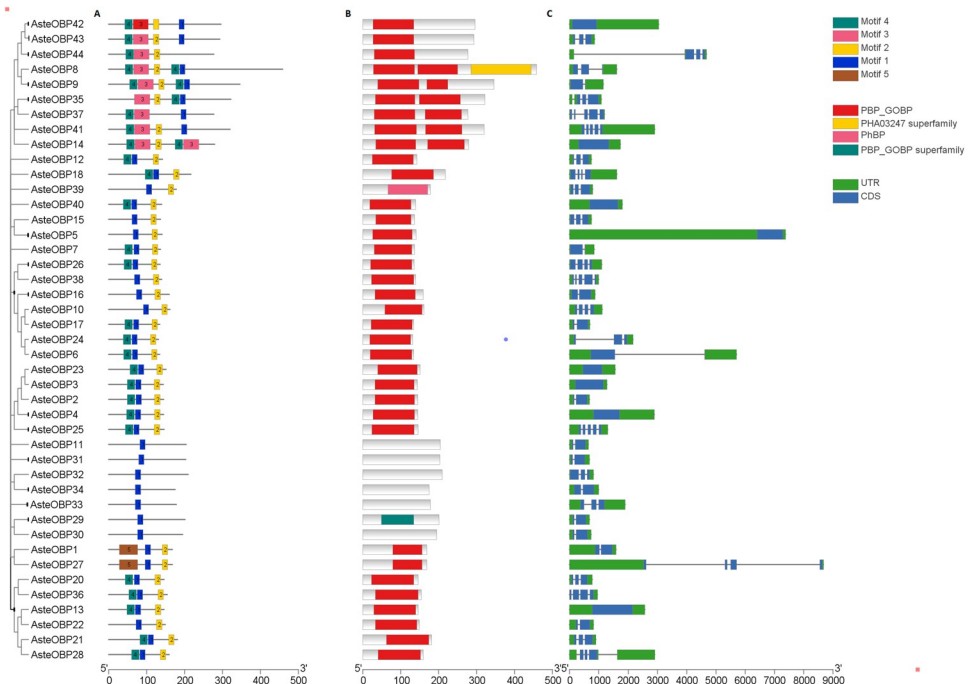

**Fig 9. Gene structure of the OBPs.** Conserved MEME motifs (A), Pfam domains (B) and UTR/CDS (C) are shown. Conserved motifs are present in all the OBPs along with the gene structure. Presence of UTRs and CDS is shown in genes.

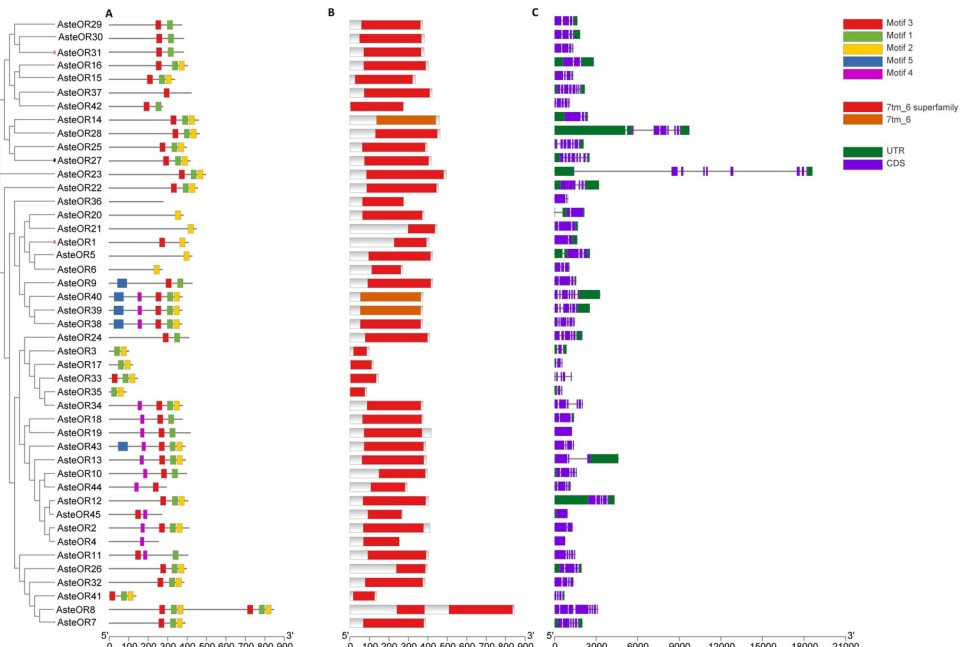

**Fig 10. Gene structure of the ORs.** Conserved MEME motifs (A), Pfam domains (B) and UTR/CDS (C) are represented in the figure. Structural domain 7tm_6 of the odorant receptors is present in all the ORs. Whereas some sequences lack the conserved motifs due to partial sequence. UTRs/CDS along with intronic regions are also represented.

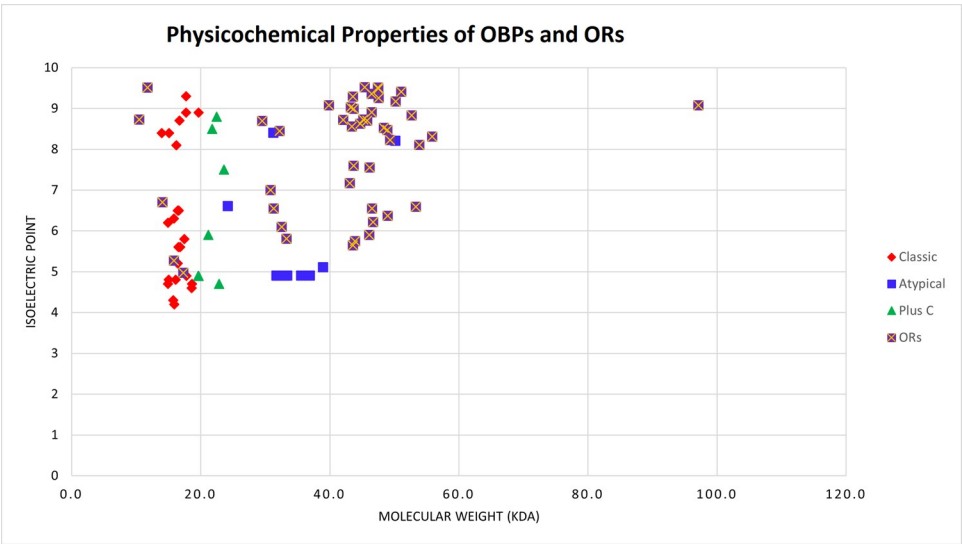

**Fig 11. Physicochemical properties of OBPs and ORs.** Molecular weight and isoelectric point of the OBPs and ORs are shown as these are important indicators of the OBPs and ORs functions. Atypical OBPs had highest molecular weight among other classes whereas ORs have higher molecular weight than OBPs.

to *D. melanogaster* [57]. Syntenic analysis showed more homologous OBPs and ORs between *An. stephensi* and *An. gambiae* genomes. Whereas less syntenic regions of *An. stephensi* OBPs and ORs are predicted with the OBPs and ORs of *Ae. aegypti*. None of the OBPs and ORs of *An. stephensi* shared homology with the *D. melanogaster* OBPs and ORs, syntenic analysis revealed. Similarly, in *Ae. aegypti*, OBPs were more related to *An. gambiae* than *D. melanogaster*. It would have been interesting to compare *An. stephensi*, a hematophagous species, with a non-hematophagous mosquito species but genome assemblies of the later, such as like *Toxorhynchites splendens* are currently yet to be sequenced.

PBP/GOBP domain (Pfam ID: PF01395) is conserved in all the OBPs despite of their subclasses. Like OBPs of other insects, classic and atypical OBPs of *An. stephensi* had significant similarity with PBP/GOBP domain and conserved motifs as well. Whereas plus-C OBPs had insignificant similarity to PBP/GOBP domain even though conserved plus-C OBP motif is present in these OBPs as checked with multiple sequence alignment. Similarly, ORs contain 7tm_6 domain (Pfam ID: PF02949) that is a transmembrane protein domain having 7 helices [54,58].

Physicochemical analysis of the OBPs predicted that atypical OBPs have the highest molecular weight in *An. stephensi* that is followed by the plus-C and classic OBPs. Similar results have been found in the OBPs of *Ae. aegypti* and *An. gambiae* as classic OBPs had molecular weight less than 15.5 kDa [47,49]. Whereas atypical OBPs of *Ae. aegypti* and *An. stephensi* had molecular weight between 27 and 38 kDa along with plus-C OBPs having molecular weight between 25–35 kDa [47,49]. However, ORs are high molecular weight proteins as their molecular weight varies from 30 kDa to 97 kDa in *An. stephensi*. The isoelectric point of the OBPs of dipteran species is found between 4 and 10 as is the case with *Ae. aegypti* OBPs [26]. Similarly, pI was between 4 and 10 for the *An. stephensi* OBPs. ORs also had pI ranged between 4 and 10 but it has not reported previously in any insect species.

With advancements in computation biology, new techniques are being devised for the functional role of the OBPs. New attractants and repellents are being discovered on the basis of molecular docking, quantitative structure-activity-relationship (QSAR), and molecular

dynamic simulations [59,60]. QSAR has been used extensively for the identification of new repellents and for the discovery of novel attractants [61,62]. Similarly, molecular docking is a fast and reliable method to screen multiple ligands for the identification of OBP-semiochemicals interactions [63]. Molecular simulations helps in the retention of ligand-receptor interactions for an extended period to induce behavioral response, thus helping to predict novel attractants [64,65]. Identification of OBPs is the first step towards the computational biology-based discovery of the novel attractants and repellents. It will not only be cost efficient but also time efficient for the *in-silico* screening of large chemical libraries. So, the identification of the OBPs and ORs of *An. stephensi*, given their vital role in olfaction process, will help in further understand mosquito olfactory system. It will help in the identification of novel attractants and repellents to control human malaria vector, *An. stephensi*.

## Conclusion

Odorant Binding Proteins (OBPs) are the first responder in the insect olfactory mechanism delivering the semiochemicals to the Odorant Receptors (ORs) in insects including *An. stephensi*. Total of the 44 OBPs and 45 ORs have been identified in the *An. stephensi* by this study. OBPs were further classified into the classic (27), atypical (10), and plus-C OBPs (7) based on the presence of the conserved motifs. Phylogenetic analysis revealed close relationship of OBPs and ORs of *An. stephensi* with *An. gambiae*. However, very few OBPs and ORs were related to *Ae. aegypti* OBPs and ORs but no relationship was established with *D. melanogaster* OBPs and ORs. Syntenic analysis showed close homology with *An. gambiae* OBPs. Physicochemical properties predicted the molecular weight and isoelectric point of the OBPs and ORs within previously reported range of OBPs and ORs. This study revealed the OBPs and ORs in the *Anopheles stephensi* which can be further characterized by NMR and crystallographic studies and help in the identification of novel compounds to control the spread of *An. stephensi*.

## Supporting information

**S1 Data. Nucleotide sequences of identified 44 AsteOBP genes in FASTA format.**
(TXT)

**S2 Data. Protein sequences of identified 44 AsteOBPs in FASTA format.**
(TXT)

**S3 Data. Nucleotide sequences of identified 45 AsteOR genes in FASTA format.**
(TXT)

**S4 Data. Protein sequences of identified 45 AsteORs in FASTA format.**
(TXT)

**S1 Table. Nucleotide length of the open reading frame (ORF) and protein length in the OBPs have been provided along with their complete and partial status.** ORFs were predicted by ORF Finder. Similarly, OBP's names and accessions have also been given.
(DOCX)

**S2 Table. Nucleotide length of the open reading frame (ORF) and protein length in the ORs have been provided along with their complete and partial status.** ORFs were predicted by ORF Finder. Similarly, OR's names and accessions have also been given.
(DOCX)

## Author Contributions

**Conceptualization:** Muhammad Faraz Bhatti.

**Data curation:** Zeeshan Zafar, Sidra Fatima.

**Formal analysis:** Zeeshan Zafar, Sidra Fatima.

**Investigation:** Zeeshan Zafar, Sidra Fatima.

**Methodology:** Zeeshan Zafar, Sidra Fatima.

**Software:** Zeeshan Zafar, Sidra Fatima.

**Supervision:** Muhammad Faraz Bhatti, Tariq M. Butt.

**Validation:** Zeeshan Zafar, Sidra Fatima.

**Visualization:** Zeeshan Zafar, Sidra Fatima.

**Writing – original draft:** Zeeshan Zafar, Sidra Fatima.

**Writing – review & editing:** Muhammad Faraz Bhatti, Farooq A. Shah, Zack Saud, Tariq M. Butt.

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
