## [Decision Letter · Decision Letter 0]

12 Oct 2021

PONE-D-21-25203Odorant Binding Proteins (OBPs) and Odorant Receptors (ORs) of Anopheles stephensi: Identification and Comparative InsightsPLOS ONE

Dear Dr. Bhatti,

Thank you for submitting your manuscript to PLOS ONE. After careful consideration, we feel that it has merit but does not fully meet PLOS ONE’s publication criteria as it currently stands. Therefore, we invite you to submit a revised version of the manuscript that addresses the points raised during the review process.

This is an important study as it describes proteins involved in odorant reception in an important malaria vector.  However both reviewers suggest ways in which the presentation could be improved including enhancement of the figures, language and an updating of the references.  The authors should respond to these comments in an effort to improve their submission. 

We look forward to receiving your revised manuscript.

Kind regards,

Joseph Clifton Dickens

Academic Editor

PLOS ONE

Journal Requirements:

Additional Editor Comments (if provided):

This is an important study as it describes proteins involved in odorant reception in an important malaria vector. However both reviewers suggest ways in which the presentation could be improved including enhancement of the figures, language and an updating of the references. The authors should respond to these comments in an effort to improve their submission.

Reviewers' comments:

Reviewer's Responses to Questions

**Comments to the Author**

1. Is the manuscript technically sound, and do the data support the conclusions?

Reviewer #1: Yes

Reviewer #2: Partly

2. Has the statistical analysis been performed appropriately and rigorously? 

Reviewer #1: Yes

Reviewer #2: I Don't Know

3. Have the authors made all data underlying the findings in their manuscript fully available?

Reviewer #1: Yes

Reviewer #2: No

4. Is the manuscript presented in an intelligible fashion and written in standard English?

Reviewer #1: Yes

Reviewer #2: No

5. Review Comments to the Author

Reviewer #1: The authors have compiled and analyzed amino acid sequences corresponding to all potential Odorant Receptors (ORs) and Odorant Receptor Proteins (OBPs) in Anopheles stephensi. They have used these sequences to reveal within species and comparative evolutionary relationships. The main value of this work is to serve as a complete reference for other researchers beginning functional studies related to neuroethology of this important species, which vectors malaria in severely at-risk human populations. Comparative analyses of these sequences shed light on the potential function of A. stephensi genes that may have evolved as adaptations to feeding in urban environments. The main weakness of this manuscript is the lack of context in the introduction and discussion. The datasets are quite useful as presented but need better framing as to their value to other researchers and relation to other anopheline species.

Introduction

The introduction would benefit from a few more statements describing the value of your data to other researchers, namely that it will be useful when embarking on functional characterization of the various olfactory sensitivities of this species or identifying the pathways most sensitive to effective repellents and attractants (as mentioned at the end of the discussion section). This could help identify novel control strategies in the future. Additionally, these genes serve as the basis for comparative studies that could explain the preference of certain subtypes for urban environments. It may be important to specify which sub-species was used in this study, as there are multiple morphological subtypes belonging to A. stephensi.

Methods

The methods are adequately described and well-chosen.

Results

Fig1 font is small relative to size of figure, making it hard to read.

Figures are pretty straightforward and helpful.

Interpretation of results

The first sentence of the discussion seems unnecessary since gene expression is not being measured here and many ORs are expressed at very low levels. Some sections of the discussion read more like extended results, offering little interpretation of the findings in a broader context. To be clear, this is a rich dataset, worthy of publication but in need of more consideration in the context of literature on other anopheline species. For example, the observation described in lines 310-311 is interesting, but there is no discussion of why this is relevant. There are other instances of data with no corresponding discussion of relevance in the discussion, like the paragraph at line 332.

Discussion is needed that compares the number of genes identified here with those found in anopheline species. The number of ORs identified is somewhat less than would be expected based on the number found in A. gambiae. This may be from difficulty of prediction or simply there are less present in this species. Some discussion is needed to qualify the significance of OR gene number in terms of how this gene family’s expansion/retraction may relate to physiology or behavior on an evolutionary scale. The same would be helpful for the OBPs.

Suggested edits by line:

Line 327 – reword sentence

Reviewer #2: The authors present an in-silico descriptive genomic analysis of Odorant Binding Proteins (OBPs) and Odorant Receptors (ORs) in the malaria vector mosquito Anopheles stephensi. Report of these gene families in this species is novel aside from a few studies that have characterized a couple of individual genes. The report is strictly an in silico analysis, there is a lack of any “wet lab” experimental validation. The authors provide interspecies comparisons to other dipterans both with respect to gene phylogeny and chromosomal synteny. Within-gene-family motif analysis is also examined for each gene family.

Considering that this is a major malaria vector in some parts of the world, it is surprising that there has not been a comprehensive report on olfactory genes in this species, as has been done for other mosquito disease vectors. So this manuscript would most certainly be worthy of publication, however major revisions to the text and figures are required before this manuscript would be suitable for publication. Many of the figures are low resolution. I am uncertain if this is a consequence of assembly of the images in a PDF, or of low-resolution images were initially provided by the authors, but it is difficult to fully assess especially the OR phylogeny relative to some of the claims made by the authors, so a better resolution image would be needed, as it is now, the text in this figure is too blurry to read.

The manuscript will also need to be thoroughly edited for language after all revisions have been made. There are numerous errors throughout the text. Furthermore, the language is not always clear nor well-structured. Careful revisions with respect to grammar, structure/organization and word choice needs be considered.

Description in the methodology is not always clear. Specifically, in the first section “Identification of the OBPs and ORs”, it is mentioned that proteome fasta files are examined for OBPs and ORs (Lines 96 and 114). Then on Lines 110-112 it is mentioned:

-OBPs were renamed according to their position on the chromosomes. Chromosomal positions of the OBP encoding genes were visualized…-

However, it has never been made clear in the manuscript that an An. stephensi genome has been examined. There is no citation or reference to a genome for this species anywhere, and it is clear that the authors have conducted some studies of these genes in the genome, so this basic information on the source of the genome would have to be clearly indicated in the appropriate section(s). This is also apparent in that in the supplemental materials file, in which nucleotide as well as protein sequences are presented for the ORs and OBPs of An. stephensi, however it is not at all clear where the nucleotide sequences have been derived from, considering that the authors only mention about proteome fasta files, and it is not clear where these proteome files are derived from. It is not always the case that proteomes are directly connected to sequenced genomes.

The authors mention in the introduction about the odorant receptor co-receptor (Orco), but it is not clear in Table 2 nor any of the figures dealing with the ORs (Fig 3, 6, 8 and 10) that they are reporting on the Orco orthologue for this species. Has this critically important gene been omitted in this report?

There are no statistical measurements provided for phylogenetic relationships in Figures 5 and 6, such as bootstrap values of maximum likelihood support values. These values are essential to ensure strength of relationship between different genes and nodes in the tree.

Table 1 and Table 2 both show basic information on the OBPs and ORs identified in An. stephensi, however this information as shown is not critically useful, especially the Protein Accession and Gene ID, which is more suitable to be included as a supplemental info. For a main-text table, such as these, it would be better to include other metrics such as completion status of the open reading frame (complete/incomplete) as well as the size of the protein (number of amino acids, molecular weight) and also isoelectric values. It is also common to show the geneID of the best blast hit from other species for each novel gene identified.

In fact, the isoelectric values and molecular weight values are summarized in Figure 11, but the actual result metrics are not presented anywhere in the report. At the very least, these values should be included as supplemental info, but could also be incorporated into a revised version of Tables 1 and 2.

With respect to the Isoelectric and Molecular Weight values, are only ORs and OBPs with complete open reading frames being considered? Because those are the only ones which could usefully be compared. It would be essential to clarify on this in the appropriate Materials and Methods section.

Specific minor comments for each section also follow:

Abstract:

Line 26. “from protein sequences of the An. stephensi” Again here it is not clear what is the source of these protein sequences. This needs to be clarified according to my previous comment, throughout the text and at the appropriate sections.

Line 35. “whereas only three OBPs and two ORs were related to the Ae. aegypti OBPs and ORs.”

Are you sure about that? Aside from difficulties in clearly reading the phylogenetic trees due to low figure resolution, it is not accurate to suggest that genes are related only if they are directly next to each other with most recent common ancestry on the phylogenetic tree. All genes within a subfamily may indeed be related to each other provided statistical support for the clustering, even if they are not the most closely related genes within the cluster.

Several times throughout the text there is statements about relatedness or lack of relatedness. Please ensure clarifications or corrections are made to these kinds of statements throughout the entirety of the manuscript.

Introduction

Line 72. “In An. stephensi, two OBPs have been identified but no ORs as of yet.”

This needs to be updated, as a publication this year, Speth et al., 2021, Insects, has reported on the characterization of an OR in An. stephensi.

Line 76-77 “ORco” should be written as “Orco” according to convention established by Vosshall and Hansson, 2011, Chemical Senses.

Materials and Methods

Line 133. “between the chromosomes of the two genomes”.

Which two genomes? In the appropriate methods section, it should be clearly written out which versions of which genomes are being analyzed for each species, and some kind of source information (accession number, literature reference, etc.) for each genome, in order to provide clear indication of which datasets are being analyzed.

Line 151. “were predicted were predicted” needs to be fixed.

Results

Line 169. “Renaming of ORs and chromosome”.

What is meant here by “renaming”? This is apparently the first description of ORs in this species, so how is it they are being renamed? Should say something like “Description or Classification of ORs” as is written for Table 1 on Line 161.

Line 183/Line193. “Chromosome 2 is saturated with…”

This is incorrect scientific terminology. “Saturated” has a very specific meaning in chemistry and its usage here is most certainly incorrect, in that the chromosomes probably could contain more OR/OBP genes than they already have. In this sense, they would not be saturated. Please consider another word choice for these descriptions.

Line 253-254. “None of the An. stephensi OBPs and ORs shared homologous regions with the OBPs and ORs of D. melanogaster.”

It is not clear what this is referring to. Homologous regions of the protein sequence? Or referring to genomic synteny? This should be more clearly written to accurately describe what is being compared.

Line 260. “Contained insignificant domains”.

What is an insignificant domain? If such descriptions are going to be made, they need to be clearly defined in the appropriate Materials and Methods section.

Line 287. “CELLO and WoLF PSORT predicted…” In the materials and methods section these softwares are mentioned but it is not clear how these determinations have been made. It would be more useful to indicate that identified OBPs contain Signal Peptides and that ORs contain appropriate number of transmembrane domains. There are specific softwares that analyze these features, but if that is what the indicated softwares have done, it would be important to clarify this in the Materials and Methods section.

Discussion.

Line 291-292. “OBPs and ORs are expressed at very high level in insects”.

This is not necessarily correct. ORs are not generally expressed at very high levels (with sometimes exceptions for Orco and sometimes pheromone receptors), and Some OBPs are indeed expressed at very high levels, but many of them are not. This needs to be corrected.

Line 301 and 302. “Aenasius bambawaei” and “Tomicus yunnanensis”.

What kind of species are these? Are they mosquitoes or other Dipterans? It should be indicated at the very least what Insect Order they belong to since the information being presented here is comparative.

Line 304-305. “which are less than the other two chromosomes in An. gambiae”.

Why is An. gambiae mentioned here. Shouldn’t this be referencing the other two chromosomes in An. stephensi?

Line 308-309. “Like OBPs, ORs are also concentrated on chromosome 3.”

This statement is logically inconsistent. On Line 303, it is mentioned that most of the OBP genes are localized on chromosome 2, not chromosome 3. So this is not a correct “like…also” statement of logic, since the patterns being compared are different. Consider to re-word this.

Line 318-319. “Likewise, Culex quinquefasciatus OBPs and ORs showed the closer relationship with OBPs and ORs of Ae. aegypti”

Closer compared to what? What exactly is the reference point? An. gambiae? D. melanogaster? Some other species?

Line 328/329. Again here with the “significant PGP domain…insignificant PGP-OBP domain”.

Significance is usually indicated to refer to statistical significance. So it is not clear what is meant by significance versus insignificance in this sense.

Line 333-334, Line 335-336. Here it says “average molecular weight of the OBPs is 18 kDA” first time and then “Average molecular weight of the OBPs is also reported to be around 50 kDA the second time.

Cannot be correct for both as written. Do you mean to say ORs for the second instance?

Figures and Tables

Figure 6, in-figure legend it says OBPs, but this figure is depicting phylogeny of the ORs. This needs to be corrected.

Figure 9B. In the in-figure legend it shows a PFAM domain indicated as PBP/GOBP. However, the PBP/GOBP subfamily of OBPs is known to be specific only to Lepidopterans, however this report is focused on a Dipteran mosquito. So it is not clear how a PBP/GOBP domain would be relevant to OBPs of this species. It is cautioned against relying too much on PFAM domains due to inaccuracies such as this. It gives the impression that all of these proteins are members of the PBP/GOBP subfamily, which they are not.

6. PLOS authors have the option to publish the peer review history of their article (what does this mean?). If published, this will include your full peer review and any attached files.

Reviewer #1: No

Reviewer #2: **Yes: **William B. Walker III

---

## [Author Response · Author response to Decision Letter 0]

25 Nov 2021

We have made all the changes recommended by reviewers. I hope our manuscript will be considered for potential publication in this journal.

Profound regards

---

## [Decision Letter · Decision Letter 1]

4 Jan 2022

PONE-D-21-25203R1Odorant Binding Proteins (OBPs) and Odorant Receptors (ORs) of Anopheles stephensi: Identification and Comparative InsightsPLOS ONE

Dear Dr. Bhatti,

Thank you for submitting your manuscript to PLOS ONE. After careful consideration, we feel that it has merit but does not fully meet PLOS ONE’s publication criteria as it currently stands. Therefore, we invite you to submit a revised version of the manuscript that addresses the points raised during the review process.

While the manuscript has been substantially improved, one of the reviewers recommends additional modifications. Please respond to the reviewer's comments.

We look forward to receiving your revised manuscript.

Kind regards,

Joseph Clifton Dickens

Academic Editor

PLOS ONE

Journal Requirements:

Additional Editor Comments:

While the manuscript has been substantially improved, one of the reviewers recommends additional modifications. Please respond to the reviewer's comments.

Reviewers' comments:

Reviewer's Responses to Questions

**Comments to the Author**

1. If the authors have adequately addressed your comments raised in a previous round of review and you feel that this manuscript is now acceptable for publication, you may indicate that here to bypass the “Comments to the Author” section, enter your conflict of interest statement in the “Confidential to Editor” section, and submit your "Accept" recommendation.

Reviewer #1: All comments have been addressed

Reviewer #2: (No Response)

2. Is the manuscript technically sound, and do the data support the conclusions?

Reviewer #1: Yes

Reviewer #2: Yes

3. Has the statistical analysis been performed appropriately and rigorously? 

Reviewer #1: Yes

Reviewer #2: N/A

4. Have the authors made all data underlying the findings in their manuscript fully available?

Reviewer #1: Yes

Reviewer #2: Yes

5. Is the manuscript presented in an intelligible fashion and written in standard English?

Reviewer #1: Yes

Reviewer #2: Yes

6. Review Comments to the Author

Reviewer #1: The revised manuscript is much more precise and the figures/tables greatly improved. All reviewer comments have been addressed.

Reviewer #2: The authors have substantially improved the manuscript, however some minor revisions are requested before this manuscript can be approved for publication. Importantly, greater clarity of documentation in some of the figure legends is required. Any symbolics in the figures should be clearly defined in the figure legends where appropriate.

1. In Figures 5 and 6, the authors have added bootstrap values. In the figure legends for these figures, it should be indicated that these numbers represent bootstrap values. In these figures some of the sequence IDs are different colors, including those that are in black font. The representations of the different colors should also be indicated in the appropriate figure legends. Additionally, the presentation of the bootstrap values in these figures is somewhat disorganized, sometimes with the numbers being overlapping and obscured by the phylogenetic relationship lines. This should be avoided, and numbers should be positioned such that they are not overlapping with other elements of the figure and it should be clear which phylogenetic relationships they are referring to.

2. In Figures 7 and 8, it is not clear to the reader what the grey synteny lines (figures 7 and 8) represent relative to the green (figure 7), blue (figure 7) or red ones (figure 8). These representations should be clarified in the appropriate figure legends

3. It was previously requested for the authors to update the tables to include ORF completion status for each OR/OBP as well as the size of the ORF, in terms of number of amino acids. This information is typically standard to include in these kinds of reports, though it has not been provided here yet. It is again requested to provide this information, either as an expansion of tables 1 and 2, appended to Figures 9 and 10, or else included as supplementary material.

4. On lines 75,76, it is mentioned that “ORs have been previously identified based on in situ hybridization using RNA probes and transcriptomic data from different organisms”. However, it is also the case that ORs have been previously identified through analysis of genomic data, as with An. gambiae and numerous other species. This approach should also be mentioned here.

5. In the first paragraph of the Discussion, from Lines 308 to 323, total number of OBPs and ORs are presented. Number of OBPs are compared to total numbers from An. gambiae, Ae. aegypti, and D. melanogaster (Lines 310-311), however a similar comparison of ORs between An. stephensi and these three species is not included towards the end of this paragraph. It would seem relevant to also compare total number of ORs across these four species.

6. On Line 377-378, “Molecular simulations helps in the retention of ligand-receptor interactions for an extended period to induce behavioral response, thus helping to identify novel attractants.” This statement is unclear. It is unclear what is meant how molecular simulations help in the retention of ligand-receptor interactions? Do the authors mean to say that the molecular simulations help to identify/predict extended ligand-receptor interactions that are retained sufficiently long enough to possibly induce behavioral response? Or is something else meant? As written it is not clear how simulations could help in retention of anything.

7. PLOS authors have the option to publish the peer review history of their article (what does this mean?). If published, this will include your full peer review and any attached files.

Reviewer #1: No

Reviewer #2: **Yes: **William B. Walker

---

## [Author Response · Author response to Decision Letter 1]

17 Jan 2022

Dear Reviewer,

All modifications have been incorporated in the manuscript as per your suggestion.

Thanks

---

## [Decision Letter · Decision Letter 2]

3 Feb 2022

PONE-D-21-25203R2Odorant Binding Proteins (OBPs) and Odorant Receptors (ORs) of Anopheles stephensi: Identification and Comparative InsightsPLOS ONE

Dear Dr. Bhatti,

Thank you for submitting your manuscript to PLOS ONE. After careful consideration, we feel that it has merit but does not fully meet PLOS ONE’s publication criteria as it currently stands. Therefore, we invite you to submit a revised version of the manuscript that addresses the points raised during the review process.

Please respond to the reviewer's comments, especially those regarding Table S2. 

We look forward to receiving your revised manuscript.

Kind regards,

Joseph Clifton Dickens

Academic Editor

PLOS ONE

Journal Requirements:

Additional Editor Comments:

Please respond to the reviewer's comments, especially those regarding Table S2.

Reviewers' comments:

Reviewer's Responses to Questions

**Comments to the Author**

1. If the authors have adequately addressed your comments raised in a previous round of review and you feel that this manuscript is now acceptable for publication, you may indicate that here to bypass the “Comments to the Author” section, enter your conflict of interest statement in the “Confidential to Editor” section, and submit your "Accept" recommendation.

Reviewer #2: (No Response)

2. Is the manuscript technically sound, and do the data support the conclusions?

Reviewer #2: Yes

3. Has the statistical analysis been performed appropriately and rigorously? 

Reviewer #2: N/A

4. Have the authors made all data underlying the findings in their manuscript fully available?

Reviewer #2: Yes

5. Is the manuscript presented in an intelligible fashion and written in standard English?

Reviewer #2: Yes

6. Review Comments to the Author

Reviewer #2: All previous concerns have been addressed, however two very minor issues remain that need to be addressed.

One is a typo found on line 272 of the untracked version. There should be a space in between "homology" and "with". In present version it is written as one word.

More substantially, in the new Supplementary Tables, specifically the Table S2 with the ORs, the complete/incomplete status for the ORs seems not to be filled in (as was the case for OBPs in Table S1), for column 3. This is critical information to include because it appears several of the ORs are incomplete. Also in the figure legend for Table S2, it is written that "Similarly, OBP's names and accessions have also be given". This should be changed to indicate that it is the OR's names.

7. PLOS authors have the option to publish the peer review history of their article (what does this mean?). If published, this will include your full peer review and any attached files.

Reviewer #2: **Yes: **William B Walker

---

## [Author Response · Author response to Decision Letter 2]

15 Feb 2022

Dear Editor,

All changes have been incorporated in the manuscript as per reviewer's suggestion.

Regards,

---

## [Editor Report · Decision Letter 3]

10 Mar 2022

Odorant Binding Proteins (OBPs) and Odorant Receptors (ORs) of Anopheles stephensi: Identification and Comparative Insights

PONE-D-21-25203R3

Dear Dr. Bhatti,

We’re pleased to inform you that your manuscript has been judged scientifically suitable for publication and will be formally accepted for publication once it meets all outstanding technical requirements.

Kind regards,

Joseph Clifton Dickens

Academic Editor

PLOS ONE
---

## [Editor Report · Acceptance letter]

14 Mar 2022

PONE-D-21-25203R3 

Odorant Binding Proteins (OBPs) and Odorant Receptors (ORs) of *Anopheles stephensi*: Identification and Comparative Insights 

Dear Dr. Bhatti:

I'm pleased to inform you that your manuscript has been deemed suitable for publication in PLOS ONE. Congratulations! Your manuscript is now with our production department. 

Kind regards, 

on behalf of

Dr. Joseph Clifton Dickens 

Academic Editor

PLOS ONE